# How Do Existing Organizational Theories Help in Understanding the Responses of Food Companies for Reducing Food Waste?

Ramakrishnan Ramanathan [1,*] , Usha Ramanathan [1] , Katarzyna Pelc [2] and Imke Hermens [3]

1   Department of Management, College of Business Administration, University of Sharjah, Sharjah P.O. Box 27272, United Arab Emirates; uramanathan@sharjah.ac.ae
2   Business and Management Research Institute, Business School, University of Bedfordshire, Luton LU1 3JU, UK; katarzyna.pelc@beds.ac.uk
3   Whysor BV, Brandemolen 65, 5944 ND Arcen, The Netherlands; imke@whysor.com
*   Correspondence: rramanathan@sharjah.ac.ae; Tel.: +971-6-5050579

**Abstract:** Food waste is a serious global problem. Efforts to reduce food waste are closely linked to the concepts of circular economy and sustainability. Though food organizations across the world are making efforts to reduce waste in their supply chains, there is currently no theoretical explanation that would underpin the responses of food companies in reducing food waste. Based on interactions with food companies over a nearly 5-year period, we explore the applicability of some well-known and not so well-known organizational theories in the operations management literature to underpin the observed responses of companies in reducing food waste. This paper is one of the first attempts to study food waste from an operations and supply chains point of view, especially from the lens of existing theories in the operations management literature and newer sustainability theories borrowed from other disciplines. Our research findings not only show that existing organizational theories and societal theories can help explain the motivations of firms engaging in food waste reduction, but also call for more research that could help explain some interesting observations that are not apparent when existing theories are used. This paper contributes to the UN's Sustainable Development Goals 1, 2 and 12.

**Keywords:** food supply chains; food waste; circular economy; organizational theories

## 1. Introduction

Food waste is a serious global problem. It has close links with the concepts of circular economy (CE) and sustainability. From a CE point of view, food waste is a kind of waste that needs attention in terms of the 4Rs, namely reduce, reuse, recycle, and recover [1]. Waste prevention is an integral part of CE approaches [2,3]. In terms of sustainability, food waste has economic, environmental, and social implications. In this sense, saving food waste contributes to several of the UN's Sustainable Development Goals.

From an operations management (OM) point of view, food waste can be reduced or eliminated via productivity improvement and lean mechanisms. The food waste sector faces huge challenges in their supply chains [4]. Any food waste that is unavoidable can then be reused or recycled in a suitable way to complete the CE cycle before the food quality deteriorates. Several initiatives have been considered to reduce and avoid food waste. The first objective of this paper is to understand the nature of food waste and its impact across food supply chains (FSCs).

While food waste is generated at various levels and at different stages, in this study, we focus on food waste occurring in FSCs. Due to the focus on reducing food waste and using circular economy principles, we use the term circular food supply chains (CFSCs) in this paper and look at opportunities for applying the 4Rs to food waste in CFSCs. A detailed

review of the food waste literature shows that most studies are practice-oriented and efforts to understand food waste practices from a theoretical point of view is missing. This is the primary research gap that this paper aims to address. Based on an extensive interaction with food companies as part of a large European research project [5], we examine whether popular existing theories in the OM discipline (e.g., [6,7]) can help us to understand the behavior of food companies and their supply chain partners. Accordingly, the remaining objectives of this research are (ii) to identify suitable theories (that are popular in the OM discipline and other theories borrowed from other disciplines) that are relevant for reducing food waste in CFSCs, (iii) to examine how these theories can help understand the observed behavior of food companies, and (iv) to identify the scope for future research in explaining certain interesting observations that are not apparent when existing theories are used. The main research question addressed in this research to fill the above research gap is as follows: how can existing organizational theories be used to explain the observed behavior of food companies in reducing food waste?

Our research will contribute to the literature in several ways. For the first time, the issue of food waste will be studied from an OM point of view, focusing on the nature of food waste and its impact across food supply chains. A review of existing OM theories and newer theories borrowed from other disciplines that are relevant for food waste and sustainability issues in CFSCs is another novel contribution of this paper. Another novel contribution is the examination of how these theories can be used to explain certain observed behaviors of food companies. Finally, we also support future researchers by highlighting the scope for future research; specifically, we focus on some of the behaviors of food companies that need to be explored in further research and beyond existing theories. Thus, our research contributes to the OM literature, CE literature, and sustainability literature.

The paper is organized as follows. The next section will provide a review of the relevant literature. Section 3 will elaborate on the interactions with agribusinesses and opportunities for theory building. The European Union has been leading efforts on CE and hence this section will provide an overview of the CE implications of food waste in the EU context. Section 4 will review theories that are relevant to understand food waste issues in CFSCs. A number of organizational theories commonly used in the OM literature will be presented in this section. In addition, we will try to borrow interesting theories from other disciplines and study how they can be used to understand issues related to food waste in CFSCs and sustainability. Section 5 will provide more discussions and the last section will provide our conclusions.

## 2. Literature Review

The term circular economy (CE) is generally defined as the practices aimed at maximizing resource efficiency in organizations [8]. Geissdoerfer et al. [9] defines it as "a regenerative system in which resource input and waste, emission, and energy leakage are minimised by slowing, closing, and narrowing material and energy loops. This can be achieved through long-lasting design, maintenance, repair, reuse, remanufacturing, refurbishing, and recycling." (Page 759). By focusing on resource circularity and optimization, CE practices contribute to increasing productivity [10,11].

The European Commission has pioneered the ideas of CE in its action plan [12], where it highlights three areas for a sustainable policy framework—designing sustainable products, empowering consumers, and implementing circularity principles in production processes. In their action plan, among other targets, they have committed to targeted food waste reduction. A CE perspective will identify opportunities that extend a product's own life cycle (e.g., via product repair), the life of its constituent parts (e.g., refurbishing or remanufacturing), or find use for the materials in the product at the end of its life cycle (e.g., recycling). From a CE point of view, waste prevention is an integral part of CE approaches [2] and needs attention in terms of the 4Rs, namely reduce, reuse, recycle, and recover [1].

The literature on CE usually focuses on business models for achieving the desired CE outcomes (e.g., Ref. [13]). Using a multiple case study approach, Vermunt et al. [1] link the 4R framework (reduce, reuse, recycle, and recover) of the CE with important CE business models—the product-as-a-service model, product life extension model, resource recovery model, circular supplies model, and hybrid models. They observe that supply chain-related barriers are not prevalent in product-as-a-service business models. In a similar study, De Angelis and Feola [14] have used a single case study approach to underline the salient characteristics of circular economy based on the ReSOLVE (regenerate, share, optimize, loop, virtualize, and exchange) framework of the Ellen MacArthur Foundation.

There is a debate in the literature about whether the concepts of CE and sustainability lead to the same level of sustainable development [15,16]. These authors highlight that CE prioritizes economic systems from an environmental point of view [12], with not much emphasis on social sustainability, while the Triple-Bottom-Line approach to sustainability will yield equal importance to all the three—economic, environmental, and social—pillars of sustainability. Accordingly, they and other similar authors [9,17] feel that CE business models (which are defined as the way in which CE principles are embedded in the value propositions in value chains) may not contribute much to social sustainability.

## 2.1. Circular Economy and Food Waste

As per WRAP [18], nearly one-third of produced food is lost or wasted. This provides an adequate background for applying CE principles in the food industry. When CE ideas are applied to the food industry, the effort is to reduce, recycle, or reuse food waste, or recover value from food waste that cannot be either recycled or reused. Reducing food waste improves the financial bottom-line for food companies and increases food availability with societal benefits. Food waste that ends up in landfills emits significant greenhouse gases and hence reducing food waste has significant environmental benefits. Thus, a circular economy business model aimed at zero food waste in circular food supply chains will be able to reach all the three pillars of sustainability [5]. Thus, saving food waste contributes to the UN's Sustainable Development Goal 1 (No Poverty), 2 (Zero Hunger), and 12 (Responsible Consumption and Production).

Thus, tackling food waste will help improve circularity and sustainability significantly. Food waste is further linked to various CE aspects such as reverse logistics, remanufacturing, servitization (or product–service systems), and sustainable supply chain management. Food waste reduction, like the focus of CE-based studies, can help organizations improve their environmental performance (e.g., waste reduction, pollution reduction, and improved ecological/carbon footprint), financial performance (e.g., profitability and economic efficiency), operational performance (e.g., productivity, product quality, and attractiveness), and social performance (health, employee morale, increased employment, and improved food security) [8].

Food waste can occur in multiple ways—at the upstream level by the producer at the production site, at the downstream level by consumers, and in between when food is moved along supply chains. There are huge consumer-behavior studies focusing on how to change the behavior and lifestyle of consumers to facilitate the reduction and complete elimination of food waste at the consumer level. Productivity studies at farms and food manufacturing plants are focusing on the upstream level. However, food waste in supply chains is a relatively unexplored area. While waste minimization in general has been a hot topic in sustainability research, understanding the mechanisms by which food companies reduce food waste in their supply chains is a relatively less explored topic. This finding has been confirmed by Kalmykova et al. [2], who, based on a literature review, observe that manufacturing and distributions are not widely studied in the context of CE. One reason for the relative under-exploration of food waste in supply chains could be because food waste that occurs in supply chains is generally considered as an unavoidable food loss [5].

*2.2. Food Waste in Circular Food Supply Chains*

Food waste is a global problem and has significant economic, environmental, social, and ethical implications. Nearly one-third of produced food ends up as waste [18]. It has been estimated that the EU produces around 88 million tons of food waste annually, equivalent to EUR 143 billion, highlighting the economic impacts of food waste. Food waste in other parts of the world paint an equally, if not more, bleak picture. Using a Life Cycle Analysis (LCA), it has been estimated that food waste alone is responsible for 8–10% of global GHG emissions.

The EU has committed to halving food waste by 2030. Target 12.3 of the UN's Sustainable Development Goals has called for halving global food waste by 2030. Several research studies have been carried out with a view to achieving these ambitious targets. For example, research studies are being conducted about when, where, and how much food waste occurs (e.g., Ref. [18]).

Based on the work from a project named FUSIONS, Parry et al. [19] have stressed the importance of preventing food waste in the first place. As per their calculations, the redistribution of food to people before it becomes waste will save 3090 kg of $CO_2$ equivalent per ton of food waste. This prevention strategy is the best strategy to fight food waste and associated greenhouse gas emissions. The calculations from their report provide very valuable information about options for treating food waste and can be linked to the 4R principles of CE. Thus, redistributing food to people before the food becomes waste is the best option, as it has the potential for saving a very high level of carbon emissions. Converting the food to animal feed is the next best option, saving 220 kg $CO_2$ equivalent per ton of food waste. Sending food waste to landfills is the least preferred option, as this will generate additional GHG emissions in landfills (about 536 kg per ton of food waste).

About 20–30 percent of food waste in food businesses occur in their supply chains–when the food is being transported or stored from the production to the final consumers [19]. A part of this loss is due to improper food storage conditions—temperature, humidity, etc.—when food produce is on the move (e.g., in a truck) or in an intermediate warehouse [20–23]. The appropriate treatment (i.e., reduce, reuse, recycle, or recover) of food waste will help food supply chains move from being linear to circular, and enable them to create circular food supply chains (CFSCs). For this study, we document our interactions with firms in European CFSCs about the behavior of food supply chain companies and their motivations to reduce food waste. We briefly discuss the food companies in the next section.

### 3. Interactions with Food Businesses and Opportunities for Theory Building

Thanks to generous funding for a large European project [5], we had close interactions with several food companies in the UK and EU for over nearly five years on our quest to support reductions in food waste. The details are given in Table 1 below.

**Table 1.** The companies involved in reducing food waste. Source: Ref. [5].

|  | Description | Country |
|---|---|---|
| 1. | Food processing in an abattoir | UK |
| 2. | Food processing in an abattoir | Republic of Ireland |
| 3. | Food storage in a frozen food company | UK |
| 4. | Milk transportation | UK |
| 5. | Food transport | UK |
| 6. | Food transport | The Netherlands |
| 7. | Food storage and transport in multiple stages of the supply chain | Luxembourg |
| 8. | Food storage and transport in multiple stages of the supply chain | Germany |
| 9. | Food storage | The Netherlands |
| 10. | Food processing—wine manufacturing | UK |
| 11. | Food production—raising cattle | UK |
| 12. | Food transportation | UK |
| 13. | Food production, storage, and transport | Germany |

While we were involved in interacting with several of these companies, some companies had only limited interactions for several reasons such as the COVID-19 pandemic. A detailed description of the nature of our involvement with several companies has already been published [22,23]. It is not our intention to describe our experiences working with these companies again in this paper, but rather to use some of our experiences to create links with theories in OM, CE, and sustainability.

## 4. Methodology

We have followed a rigorous methodology (Figure 1) for answering the research question and achieving the research objectives specified in Section 1. This includes a review of prominent organizational theories, identifying major themes advocated by these theories, linking our observations when working with food companies to reduce food waste with these major themes, and identifying the behaviors of firms that are not directly apparent from these theories.

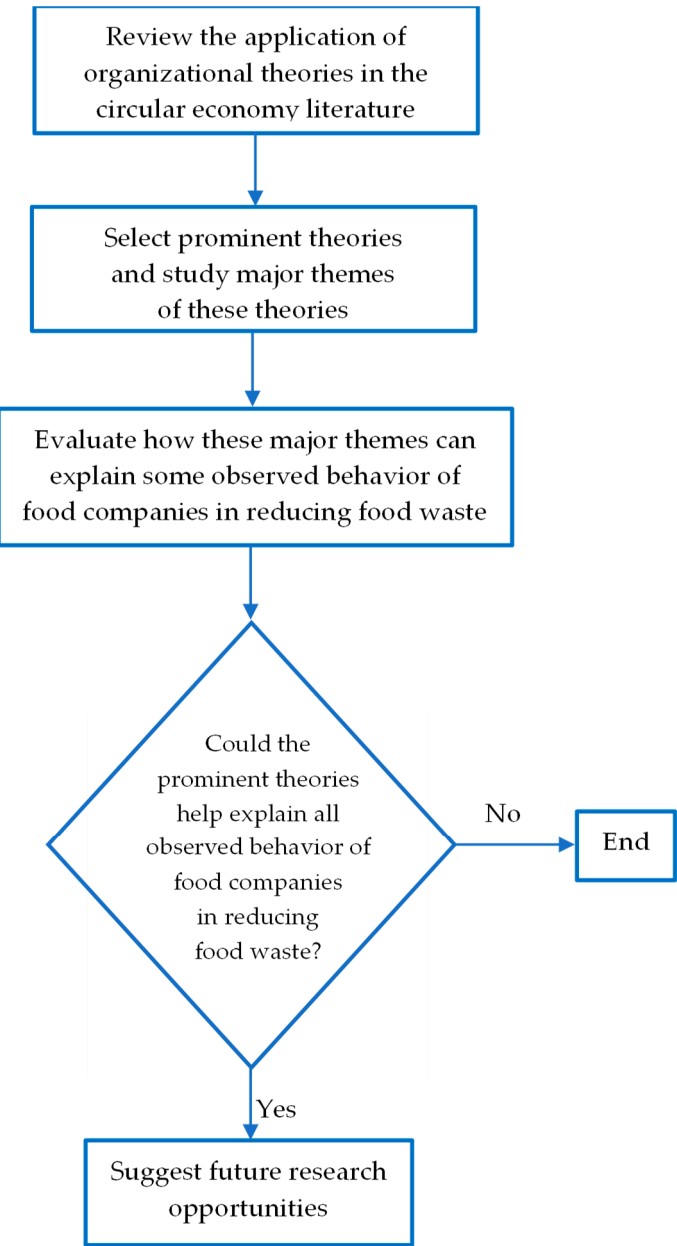

**Figure 1.** Research methodology steps.

As per these steps, we first provide a brief overview of some important theories in the next section.

## 5. Theoretical Underpinnings for Food Waste

We first provide an overview of some important theories in this section and use these theories to explain the behavior of the food companies listed in Table 1. These theories are reviewed from the CE and sustainability literature, where we highlight those theories that are used by a majority of OM researchers and those that are not well employed in OM research. As highlighted earlier, our emphasis is not to provide a detailed disposition of these theories, but rather to link important tenets of these theories to our observations about the behavior of food companies. Readers interested in understanding the theories in detail may refer to other suitable references [6,7,24]. Here, we highlight that there has been no study that has applied these theories directly to food waste practices, and hence we draw on the theories in the field of circular economy.

Various organizational theories have been employed by CE researchers to understand the behavior of firms, though it has been observed that these theoretical approaches and business models are under-researched topics when applied to circular economy [8,25]. Ref. [26] has listed a total of 15 management theories in their review of the literature on CE business models. The theories include some theories that are familiar to OM researchers (e.g., the business model theory, dynamic capabilities perspective, industrial network theory, marketing systems theory, principal-agency theory, systems theory, transaction cost theory, stakeholder theory, and institutional theory), and some that are not so familiar (e.g., eco-efficient value creation, ecological economy and social/solidarity theory, industrial ecology theory, life-cycle thinking theory, prospect theory, and the ecological modernization theory). Similarly, Sehnem et al. [8] have reviewed the literature and found a limited set of theories (including the institutional theory, stakeholder theory, resource-based theory (RBT), industrial ecology, transaction cost economics, and social network analysis) that have been applied to the CE literature. Based on a review of the literature on GSCM and CE, Liu et al. [6] have identified that twelve theories have been applied to both the disciplines, while seven additional theories have been applied only to GSCM, and eight more have been applied only to CE. Lahti et al. [27] describe in detail how six commonly used organization theories—contingency theory, TCE, RBT, path-dependence theory, and agency theory—can be used to understand the behavior of firms engaged in CE.

The major organizational theories in the CE literature include the institutional theory [28], the stakeholder theory [15], and the resource-based theory [25]. Based on a literature review, Gusmerotti et al. [29] have identified the institutional theory and the RBT theory as the most appropriate organizational theories that can explain behavior of firms engaged in circular economy practices. However, there seems to be a confusion in the development of theories in the CE literature, as many authors (e.g., Ref. [29]) seem to borrow theories from the sustainability literature directly without providing adequate attention to the distinction between sustainability and CE.

Ranta et al. [28] highlight that the institutional theory can be used to explain the drivers and barriers of CE. They focused on institutional drivers and barriers for the 3R (reduce, reuse, and recycle) principles of the CE. Drawing from Scott's institutional theory framework [30], they have used the three pillars—regulative, normative, and cultural–cognitive—to highlight the differing pressures on institutions engaging with circular economy ideas.

The stakeholder theory [31] is commonly employed in sustainability research. As CE is closely linked to sustainability, the need for considering requirements of all stakeholders has been discussed in several articles that discuss enhancing the value propositions of CE-based business models [15,32,33].

Lopez et al. [25] identify five main categories of barriers (institutional, market, organizational, behavioral, and technological) for resource efficiency in firms. Vermunt et al. [1] have prepared a slightly larger set of barriers, adding supply chain barriers, financial barriers, and knowledge barriers to the above list. For example, the lack of appropriate

partners, low availability of raw materials, higher dependence on external parties, and lack of information exchange/conflict of interests between supply chain actors are crucial supply chain-related barriers that could affect circular economy business models. These barriers support the application of the RBT to CE.

In an interesting recent development, De Angelis [14] has linked several tenets of the paradox theory with CE business models. Paradoxes exist when contradictory, yet interrelated elements occur simultaneously in a system. Several categories of paradoxical tensions (viz. learning, organizing, belonging, and performing paradoxes) for CE principles and CE value loops have been identified [14].

There has been more interest in the literature about the use of technologies in reducing food waste. For example, Li et al. [34] have reviewed the literature on the use of blockchain in food supply chains. Li et al. [35] have focused on the specific application of internet of things sensors for perishable food management. Stefanini and Vignali [36] have further highlighted how new technologies can help food companies achieve the three pillars of sustainability. Traditional innovation theories such as the innovation diffusion theory [37], the technology–organization–environment (TOE) framework [38], and the technology acceptance model (TAM) [39], including its latest versions (e.g., the unified theory of acceptance and use of technology or UTAUT [40]), are some common theories that can be associated with the use of technology for reducing food waste.

There is a general awareness in the CE literature about the importance of technologies. For example, Bressanelli et al. [41] explain how new digital technologies such as IoT and big data analytics can help support circular economy practices using a single case study. Tseng et al. [42] highlight the paucity of research articles on circular economy that exploit the power of newer digital technologies (including IoT technology, big data analytics, cyber–physical systems, cloud computing, artificial intelligence, and more) and explain that these technologies have a huge scope for the transition from the current linear economy to the more sustainable circular economy. Others have suggested OM-linked CE business models using the ReSOLVE framework of CE [26], where they stressed the value of new digital technologies for CE business models for operations, logistics, and supply chain activities.

Given the importance of technology in CE, efforts were focused on how we can best utilize technologies to achieve desired outcomes. Accordingly, theories developed in the context of technology have been employed. Using a literature survey and based on the ReSOLVE framework, Ref. [26] brings out multiple propositions linking CE with big data principles. The authors have suggested key stakeholders for each stage of the ReSOLVE framework and go on to explain how the stakeholder theory and institutional theory can form bases for understanding the links in greater detail in future studies. In fact, speculating on the directions of CE and big data in the future, they feel that multiple theories can help in future research in this direction. The theories they suggest include the resource-based theory and the dynamic capabilities theory that are more commonly used in operations and supply chain management research. In addition, they suggest borrowing newer theoretical ideas from other domains, including those in social and policy studies (e.g., ecological modernization theory) and information technology fields (e.g., technology acceptance models). In fact, there is huge scope for a rich understanding of CE from multiple theoretical frameworks. Amidst the development of OM theories, there is also a great concern with the evolution of novel theories. The dynamic nature of multiple theoretical frameworks creates avenues for the applicability and invention of new theories or the adaptation of existing theories. For example, some theories can be disproved by looking at events that are contrary to estimations [43,44].

Grover and Dresner [45] presented a theoretical model explaining how political resources could be aligned with supply chain strategies. This can be compared to the resource dependent theory or resource-based view. On a different note, another research paper [46] discussed publicness theory and supply chain integration. Sarkis et al. [7] presented a detailed literature review on green supply chain management theories to understand green

concepts from several fields. They considered complexity theory, ecological modernization theory, and information theory alongside resource theories. Since 2010, technological intervention in agribusinesses has taken a great role in business performance.

Though not yet studied in detail in the context of food waste and circular economy, we feel that the institutional entrepreneurship theory could play a significant role in explaining entrepreneurial motivations in CFSCs. One of the most influential papers on institutional entrepreneurship to date has been developed by Battilana et al. [47]. Institutional entrepreneurs are change agents, who (i) initiate divergent changes and (ii) actively participate in the implementation of these changes. Only actors who initiate divergent changes, that is, changes that break with the institutionalized template for organizing within a given institutional context, can be regarded as institutional entrepreneurs. Battilana at al. [47] list two conditions enabling institutional entrepreneurship—field-level conditions and an actor's social position. In the context of CE, institutional entrepreneurship theory can be applied to understand how some actors in agri-food supply chains break with the dominant logic/template/way of doing things and introduce a new way of doing things.

The discussion above brings out some prominent theories used in OM and CE—the stakeholder theory, institutional theory, resource-based theory, paradox theory, resource dependence theory, and institutional entrepreneurship theory. This study will focus on these theories in the next section. Another important observation is that all these theories have been discussed in the context of CE, while there has been no effort to link these theories to the motivation of CFSCs in reducing food waste. This is a significant research gap. The next section contributes to the literature by filling this important research gap.

### 5.1. Applying Theoretical Underpinnings to Study Motivations for Food Waste Reduction in CFSCs

We believe that several of the organizational theories mentioned so far in this section can be used to explain the motivation of organizations engaging in food waste in CFSCs. While all the Rs of the CE [28] are relevant for food waste, reusing and recycling are more important in terms of social, economic, and environment impacts. By reusing food before it becomes unfit for human consumption, significant carbon emissions can be reduced [19].

In the paragraphs below, we utilize our interactions with food companies to explain the applicability of common organizational theories for reducing food waste.

#### 5.1.1. The Stakeholder Theory

Some principles on the application of the stakeholder theory to food waste can be borrowed from the sustainability literature. This theory can be used to explain some interesting behavior of food businesses engaging in the reduction of food waste in their supply chains. Several stakeholders are influencing food companies to reduce food waste. Based on our interactions with food companies, we identified that the government, via legislation, is one of the most influential stakeholders. Governmental regulations play a strong role here. For example, regulatory systems can force companies to comply with the regulations and hence help them reduce food waste. This is apparent in the agri-food industry, with European directives such as the Hazard Analysis & Critical Control Point (HACCP) EU Directive. This regulation, introduced in the EU in the 1990s and modified in subsequent years, expects EU food business operators to put in place, implement, and maintain a permanent procedure or procedures based on the HACCP principles. Without an appropriate plan to avoid hazards such as the contamination of food with bacteria, fungi, viruses, and parasites, food items may cause several food-borne illnesses in consumers. The plan could include, for example, maintaining the correct atmospheric conditions (temperature, humidity, etc.) that would keep the shelf life of food long enough, which in turn would avoid them becoming waste quickly.

These regulatory pressures can act as barriers if companies perceive that these regulations are not effectively enforced by local/regional governments. Other stakeholders are also important for reducing food waste. Top management commitment and commitment

from employees play a strong role in reducing food waste. Other downstream supply chain partners, by virtue of their position as customers, also exert pressures on reducing food waste.

### 5.1.2. The Institutional Theory

Some tenets of the institutional theory can be used to explain the behavior of the food companies engaged in technology demonstrations. The three pillars of the institutional theory [28,30] can be applied to understand why and how firms in agri-food supply chains can engage in actions to reduce food waste. The three pillars—regulative, normative, and cultural–cognitive—of this theory can provide motivations and inhibitors for reducing food waste. Reducing food waste can be internalized by food companies using the normative pillar of the institutional theory. This can be implemented if all stakeholders of a firm believe that disposing food waste in landfills is less preferable to, for example, donating to charities. This internalization is important to motivate companies when they perceive that the costs of technology investments in reducing food waste are larger than the market value of avoiding food waste. Explicit associations with established food charities can be a good motivator for the normative pillar. The normative pillar was manifested in the companies listed in Table 1 when they prioritized their own survival and were hesitant to engage in the innovative activities of the project, even though they knew that working on the project would benefit them in due course. We experienced another manifestation of the normative pillar when some of the food companies joined the project for the green image it generated. The cultural–cognitive pillar of the institutional theory represents practices that involve mostly unconsciously adopted decisions. For example, there is in general a higher emphasis on reducing food waste and adopting sustainable food practices in modern days compared to a few decades earlier, which can explain why all food companies are putting more and more efforts, even when some of them are not required by law, to reduce food waste in their supply chains.

The institutional theory has also been used to explain the barriers to the implementation of CE in organizations. Lopez et al. [25] identify five main categories of barriers (institutional, market, organizational, behavioral, and technological) for resource efficiency in firms. Of these, we had opportunities to observe three—organizational, behavioral, and technological—categories. Specifically, we observed that a willingness of firms in terms of favorable changes in behavioral and organizational efforts are critical for the right application of technology and the maximum reduction of food waste. Vermunt et al. [1] have prepared a slightly larger set of barriers, adding supply chain barriers, financial barriers, and knowledge barriers to the above list. For example, the lack of appropriate partners, low availability of raw materials, higher dependence on external parties, and lack of information exchange/conflict of interests between supply chain actors are crucial supply chain-related barriers that could affect circular economy business models. This is equally true for reducing food waste in food supply chains. In fact, during our interactions with food companies, we experienced that generally negative perceptions of IT projects were a significant barrier to overcome.

### 5.1.3. The Resource-Based Theory (RBT) and NRBT

We believe that the resource-based theory (RBT) [48] and the natural resource-based theory (NRBT) [49] provide very good opportunities to explain the behavior of food companies engaging in food waste reduction efforts in their supply chains. The NRBT is required if we view food waste in the context of the imputed natural resources (energy, labor, soil, fertilizers, water, and more) needed to produce the food. The issue of valorizing food waste can be used from the lens of the RBT. Although the RBT has not yet been applied to the case of food waste, a related concept, called the resource-based paradigm, has been shown to help view waste in terms of resources [50]. This idea can be extrapolated further to bring the RBT into the food waste context, if we emphasize that the unique, inimitable knowledge generated in firms that have started to view waste as another resource, enables

firms to gain competitive advantages. The knowledge of the composition of what is currently termed as waste and understanding the potential utility of waste can become accumulated into an inimitable knowledge in the long run. Continuously looking for technological innovations in-house and elsewhere to valorize what is currently termed as waste in a firm can not only generate more revenues via extra sales but also can lead to a reduction in costs via reduced raw-material consumption and reduced waste disposal costs. This observation has been somewhat echoed by Ref. [2] in their discussion of the Circular Economy Strategies Database that captures 45 CE strategies (e.g., material substitution, green procurement, product labelling, eco design, re-use, recycling, extended producer responsibility, and more) in various stages (e.g., material sourcing, design, manufacturing, distribution, consumption, collection, recycling, and more) of the economy. Implementing these strategies efficiently will result in valuable and inimitable knowledge for improving the resource efficiencies of operations in businesses. Thus, understanding opportunities for CE in a business will provide the scope for a firm to gain a competitive advantage from the point of view of the RBT.

Klassen and Whybark [51] and similar researchers working on pollution prevention have used the RBT to differentiate between pollution-prevention technologies (i.e., technologies that prevent pollution from occurring) and pollution-control technologies (i.e., technologies that try to reduce the impact of pollution once it has occurred). In a similar way, the RBT can also help us to understand the implications of waste prevention vs. waste control in CE organizations. In the context of food waste, waste prevention would mean that efforts are made to avoid food waste from occurring. There are a number of strategies available to organizations to achieve this. For example, effective information sharing from supply chain partners can help produce food in the right quantity for a given purpose with little waste. Effective scheduling in food-processing industries or in the farming industry can help reduce waste. Using appropriate technologies can improve productivity and reduce waste. The main aim of our interactions with food companies is to avoid waste from occurring in the supply chain by ensuring that produce is kept in the right conditions. These strategies will help to prevent food waste from occurring. The RBT can help successful firms to mobilize their available resources to prevent waste and gain competitive advantages. By preventing waste, firms are able to reduce the cost of raw materials consumed, improve quality, reduce their waste disposal costs, and thus gain competitive advantages. Waste-control strategies are useful to limit the impact once waste has occurred. They do not result in as much of a competitive advantage compared to waste-prevention strategies. Thus, the RBT helps firms to look for opportunities to prevent waste first.

The economic advantage derived by reduced food waste translates into competitive advantages to firms. This economic angle provides another way that the RBT can be used to explain the behavior of firms in reducing food waste. One rather more interesting way that the RBT can be used to explain the motivations of agribusiness companies in reducing food waste is the quality angle. The food companies we worked with mentioned that they do not incur food waste anymore after engaging with us about the use of technologies. On their view, digital technologies primarily help them in improving the quality of their produce. Using the appropriate monitoring of quality-related variables, these food companies are more confident that their food produce will have high quality in the market. The literature on quality management [52] explains that investments in improving quality help firms in gaining competitive advantages via improved market prices and reduced waste.

### 5.1.4. The Paradox Theory

The paradox theory [53] can also help understand the motivations of firms in the food waste context. Given that food waste can be reduced via soft means (e.g., behavioral changes) or relatively hard means (e.g., using technological support), there is a learning paradox in deciding on the relative importance of these two means. To reduce food waste in food supply chains, there is a need for supply chain collaboration, but at the same

time, different supply chain partners need to maintain their identities, giving raise to both organizational and belonging paradoxes. A paradox is apparent when one needs to verify whether technological investments in reducing food waste consume more resources compared to the resources imputed to the food that is likely to be saved.

### 5.1.5. The Resource Dependence Theory (RDT) and NRDT

The resource dependence theory [54] and the natural resource dependence theory (NRDT) [55] will be relevant in the context of food waste in FSCs. The RDT emphasizes external dependence on scarce resources and the uncertainty that it creates for organizations to survive [54], while the NRDT explicitly emphasizes additional dependence on natural resources [55]. Members of a supply chain are traditionally dependent on each other for the continued success of the supply chains and hence their own survival, which explains the applicability of the RDT for CFSCs. For example, firms depend on data from other supply chain partners for making prompt business decisions and consider data sharing and security (including threats from hackers) as crucial limiting issues for their growth. The use of modern technologies exacerbates these dependencies, for example, for global connectivity. In some cases, a food producer may not be able to install gateways to transmit sensor signals from a truck if lorry drivers (which are another crucial resource in CFSCs) object to having too many transmitting devices in their cabin. As highlighted earlier, since food waste involves significant but scarce natural resources for producing food, there is a crucial dependence on natural resources too.

### 5.1.6. The Institutional Entrepreneurship Theory

As mentioned earlier, due to its ability to explain entrepreneurial motivations, the institutional entrepreneurship theory [47] could play a significant role in helping us to understand the motivations of firms in reducing food waste. Several EU policy makers have highlighted that the European Commission has only recently started to coordinate national policies about food waste in member states, and in some cases (where such policies did not exist) it has started to push member states to develop such policies. The European Commission has used its 'social position' in the field to push for a major institutional change—at both the member state and EU level—to develop policies for food waste at the national and EU levels. Investment firms have highlighted that profit was not the key decisive criterion for these firms to invest in sustainability. There is growing evidence that firms are breaking away from the dominant stereotypical ways of behaving. For example, firms are explicit in stating that profit generation is not enough anymore; the food business needs to also promote animal welfare and respect the environment and natural resources. Some of the previous discussions did highlight many additional reasons for companies to engage in the reduction of food waste in their CFSCs, including legal requirements and quality enhancement. These considerations, we think, are examples of institutional entrepreneurship—breaking with the dominant template and way of doing things.

As a summary of this section, Table 2 presents major tenets of important organizational theories and how they can be interpreted in the context of food waste reduction in circular FSCs.

**Table 2.** Elements of various organizational theories for managing food waste in circular food supply chains.

| Theory | Element | Links to Food Waste |
| --- | --- | --- |
| The stakeholder theory | Multiple stakeholders | • Regulatory stakeholder (HACCP directive)<br>• Top management commitment<br>• Employees<br>• Supply chain partners |

**Table 2.** *Cont.*

| Theory | Element | Links to Food Waste |
|---|---|---|
| The institutional theory | Regulative, normative, and cultural–cognitive pillars | • Firm belief in reducing food waste<br>• Preference for donating to food charities than sending to landfill<br>• Explicit association with established food charities<br>• Prioritizing survival to innovation during COVID-19 lockdowns<br>• Associating with the green image<br>• Voluntary initiatives on reducing food waste and sustainable food practices<br>• General lack of trust in IT projects |
| The resource-based theory and the natural resource-based theory | VRIN (valuable, rare, inimitable, and non-substitutable) resources and competitive advantage | • Wasted food is a waste of valuable imputed natural resources (energy, labor, soil, fertilizers, water, and more)<br>• Inimitable knowledge on reducing, recycling, and reusing waste, and valorizing food waste is a source of competitive advantage<br>• Efficient operations management for reducing raw material consumption for competitive advantage<br>• Waste prevention vs. waste control in CE organizations<br>• Efficient quality control for competitive advantage |
| The paradox theory | Learning, organizational, and belonging paradoxes | • Soft means (e.g., behavioral changes) vs. relatively hard means (e.g., using technological support) for reducing food waste<br>• Supply chain collaboration vs. maintaining individual identities of partners<br>• Comparing costs of technological investments in reducing food waste with the resources imputed to the saved food |
| The resource dependence theory and the natural resource dependence theory | Supply chain dependency | • Dependence on data from other supply chain partners for making prompt business decisions<br>• Data sharing and security issues<br>• Use of modern technologies exacerbates these dependencies |
| Institutional entrepreneurship | Breaking away from dominant ways of doing things | • Entrepreneurs and enterprises do not consider profit as their single motive anymore. Other considerations including social and environmental impacts are increasingly being employed in entrepreneurial decision-making. |

## 6. Discussion

As highlighted in Table 2, the six theories we have selected in the previous section can help us to understand the several observed behaviors of food companies. For example, the stakeholder theory helps to visualize the government as the regulatory stakeholder and explain how food companies approach the HACCP directive. It also helps to see

why food companies should consider the views of other stakeholders, such as the top management, employees, or supply chain partners, while making decisions on how to reduce food waste. The institutional theory has helped us to explain why food companies prioritized their own survival rather than engaging in innovations for reducing food waste during the pandemic. It also explains the general lack of trust in IT projects for reducing food waste. The resource-based theory has helped us to explain the trade-offs in food companies between investing in waste-prevention vs. waste-control options. While the waste in food supply chains can be reduced by sharing information, the issues of privacy and security has been explained using the resource dependence theory. The fact that some companies attempt to lead in the development of strategies for food waste reduction more than others has been explained using the institutional entrepreneurship theory.

In summary, we believe that there is no single theory for explaining all the behaviors of food companies. Interestingly, each theory is able to explain only a part of these behaviors. We further believe that all the six theories listed in Table 2, together, are able to explain a majority of the behaviors of food companies in reducing food waste. This might call for a proposal of a *unified theory of food waste* by bringing the features of all six theories together. This calls for interesting research by future researchers.

Many of the theories discussed in this article have been developed in the past several decades using organizational, social, economic, ethical, and sustainable viewpoints. Among their diverse approaches, all of them point towards the enhancement of the subject being considered. Two main themes, namely sustainability and circular economy, are brought together in the 21st century to visualize our green future with financial and social prosperity. Our paper has discussed the theories and their contribution to CFSCs in detail; this could lead to many future research projects in the area of supply chains and to collaborations aimed at implementing sustainability and circular economy practices with societal involvement.

In spite of the support of these theories in explaining some observed behaviors of food companies, some other observations could not be readily explained using any of the six theories above. For example, we observed a strange behavior when we attempted to install some new technologies to control and monitor storage conditions of food in trucks. Since any new piece of technology (e.g., IoT sensors) installed in trucks could also track the location of vehicles, drivers were reluctant to engage in the installations. This negates, for example, the resource dependency theory. Similarly, when we could not support a specific request of a company due to technological limitations (for example, due to a lack of the availability of a suitable low-cost technology to continuously facilitate international shipments in multiple continents), the company did not wish to engage with us further for other parts of their operations. Similarly, another company decided not to work with us after their internal restructuring. We may need newer theories or borrow from several other disciplines to explain this behavior.

We believe that the discussions in the preceding sections could provide a basis for classifying theories in the context of food waste. In addition to the prominent theories discussed in this paper, other theories can be potentially applied to understand the behavior of firms regarding food waste. They include systems theory, systems thinking theory, systems dynamics theory [56,57], the intermediary actors theory [58], social practice theory [59], perceived behavioral control/the theory of planned behavior [60], information systems theory [61], the network design theory [62], creating shared value theory [63], and more. A comprehensive view of several theories discussed above could be integrated around the theme of circular economy and circular food supply chains. Figure 2 is representative of the achievable pathways for CFSCs and global sustainability. Each of the theories from various perspectives can be brought together to visualize sustainability in the long run. We have positioned the theories that are enabling and enriching the sustainability of CFSCs under six main perspectives (refer to Figure 2). These theories can be considered as imperative to understand the concepts of sustainability in agribusiness supply chains and are positioned in the inner circle of Figure 2. These perspectives are named as dynamic and

innovative perspectives, institutional and business perspectives, economic perspectives, social networking perspectives, resources perspectives, and ecology and environment perspectives. The theories in the outer circle in Figure 2 are either based on technology usage or borrowed from other disciplines (e.g.,predator–prey theory [64], chaos theory [65,66], complexity theory [7], and the institutional entrepreneurship theory [47]). We call these theories supporting theories, as we have yet to see a large-scale application of these theories to food waste issues; however these theories could provide additional support for the cause of food waste reduction.

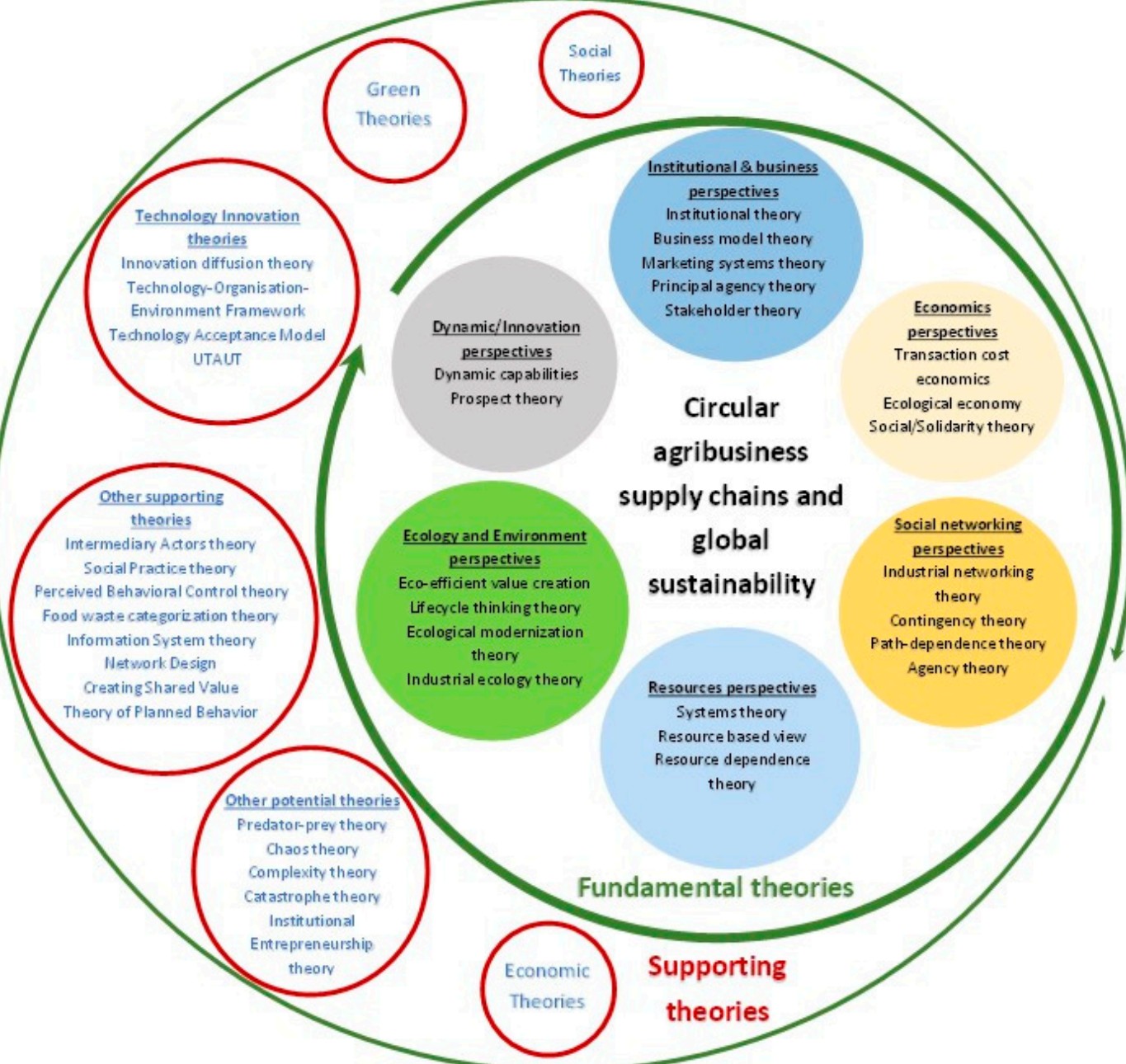

**Figure 2.** A pictorial representation of various theoretical perspectives woven around CFSCs and global sustainability principles.

Some theories have been criticized for their applications to practical matters. For example, although the RBT has been used widely in OM area, there is a series of conversations about making improvements to the theory and its applications. While Singh et al. [67]

used the theoretical lens of resource dependence theory to understand ISO 9000 for organizational green management, Hitt et al. [68] suggested RBT enhancement through the adoption of various applications. These on-going conversations clearly specify the richness of the theory and its impact in practice. We can say that OM theories, in combination with theories from other disciplines, are the backbone for the development of applications, new practices, and new methods. The concept of circular economy in agri-business is a relatively new concept that needs enrichment and support from theories across disciplines to show its real potential for economic, social, and environmental practices.

In summary, we can classify theories into two main categories: (i) fundamental theories (six perspectives given in Figure 2) of food waste that strengthen our understanding and the importance of identifying areas of potential food waste reduction, and (ii) food waste supporting theories that support actual food waste-reduction intiatives (in the outer circle of Figure 2). Using our empirical research related to the European food sector, we classify the paradox theory, the resource-based theory, institutional theory, and stakeholder theory as the fundamental theories, and we classify the institutional entrepreneurship theory and technology organization theory as the supporting theories. In simple terms, the fundamental theories will act as inputs and the supporting theories will serve as outputs for any research that considers the socio-economic and environmental aspects of waste reduction and sustainability.

## 7. Conclusions

Given the growing importance of food waste in meeting several of the UN's Sustainable Development Goals, we studied the issue of food waste from an operations management point of view in this paper. Our main research question was to understand how existing organizational theories could help to understand the observed behavior of food companies in reducing food waste. We have addressed this research question via Table 2. The research objectives have also been achieved as we have identified suitable theories and used them to understand food waste practices. With the help of the literature on circular economy, we studied circular food supply chains (CFSCs) using multiple theories. We specifically looked the stakeholder theory, institutional theory, the resource-based theory, the paradox theory, the resource dependence theory, and the institutional entrepreneurship theory in greater detail in the context of food waste.

This paper contributes to the literature on operations, supply chains, and circular economy in multiple ways. Though the importance of food waste was recognized long ago, this is the first time the issue of food waste has been studied from an OM and supply chains point of view. We reviewed existing theories related to circular economy and discussed some newer theories that may hold promise to support a better understanding of circular economy. We used at least six of these theories, for the first time, in the context of food waste.

In spite of these contributions, there are limitations to our approach in this paper. The data from our qualitative study come from a small sample of 13 companies listed in Table 1. We think that our continuous association with these 13 companies over a period of more than 4 years has yielded rich insights in explaining organizational behavior. However, data could be gathered from more companies to yield more generalizable results. Similarly, our sample focused only on companies based in North-West Europe. For better generalizations, more companies from other parts of the world could be studied. Finally, we only focused on six important theories and applied them to understand the food-waste context. However, several more theories (e.g., the agency theory, the contingency theory) can also be applied; we could not focus on them due to the limited time and space. Some exciting new theories, borrowed from the engineering literature, can also help to understand the circular economy principles, but we did not discuss them due to lack of time and space. For example, the catastrophe theory [69] has the capability to explain why, how, and when public perceptions about specific features of circular economy will change. Akin to the sudden change in the public perception of the use of plastics, public perceptions of waste streams can also quickly

change, which can be studied using the catastrophe theory. Future papers can consider this theory to support circular economy practices in greater detail. Another area of interest that we could not examine in more detail in this paper is the distinction between food-waste prevention and food-waste control. Borrowing from the pollution-prevention/control literature, we explained how the resource-based theory can help understand the relative merits of food-waste prevention and control. This issue can also be studied in more depth in future studies.

We are confident that the analysis of food waste for CFSCs will be useful to researchers engaged in theoretical studies of food waste and to policy makers engaged in food policy and circular economy.

**Author Contributions:** Conceptualization, R.R. and U.R.; formal analysis, R.R. and U.R.; funding acquisition, R.R., U.R., K.P. and I.H.; investigation, R.R., K.P. and I.H.; methodology, R.R. and U.R.; project administration, R.R. and K.P.; resources, R.R. and K.P.; writing—original draft, R.R.; writing— review and editing, U.R., K.P. and I.H. All authors have read and agreed to the published version of the manuscript.

**Funding:** This research was funded by Interreg North-West Europe (NWE831).

**Institutional Review Board Statement:** The study was conducted after gaining ethical approval (ref BMRI/Ethics/Staff/2018-19/005) from the University of Bedfordshire, UK.

**Informed Consent Statement:** Informed consent was obtained from all subjects involved in the study.

**Data Availability Statement:** The original contributions presented in the study are included in the article, further inquiries can be directed to the corresponding author.

**Acknowledgments:** This research has benefited from the insights about food waste reduction in agribusiness supply chains from the REAMIT project. We acknowledge the contribution by members of the project though we could not include all their names as authors in this article.

**Conflicts of Interest:** Author Imke Hermens was employed by the company Whysor BV. The remaining authors declare that the research was conducted in the absence of any commercial or financial relationships that could be construed as a potential conflict of interest.

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
