# Peer review of "How Do Existing Organizational Theories Help in Understanding the Responses of Food Companies for Reducing Food Waste?"

_sustainability, doi:10.3390/su16041534_

Round 1

Reviewer 1 Report

Comments and Suggestions for Authors

My comments are as follow:

·         Please, do you consider the topic original or relevant in the field? Does it address a specific gap in the field? Please, indicate it clearly in the introduction section.

·         Moreover, what is the main question addressed by your research? Please, indicate it at the end of the introduction section; and later, correlate that question with a conclusion in that section.

·         A specific improvement that you should consider regarding the methodology can be include a figure indicating the steps applied in your research.

·         Line 540, change “In” instead of “in”.

·         I suggest to improve the “Discussion section”. You include the subsection: “5.1.1. Towards a classification of theories in the context of food waste”; and then, there was not any more subsections.

·         I suggest to summarize the “conclusions section”. Additionally, are the conclusions consistent with the evidence and arguments presented in the Literature Review section, int the interactions with food businesses and opportunities for theory building section, ant the theoretical underpinnings for food waste section. Moreover, the conclusions address the main question posed in your manuscript? Please, indicate it clearly.

·         What does it add your research, to the subject, area compared with other published material? Please, indicate it clearly.

·         You stated that the “type of the paper is an article”. Nevertheless, in my point of view, the manuscript is more an “literature review”. If you accept this suggestion, please, change it accordingly.

Author Response

Please see the attached responses document.

Reviewer 2 Report

Comments and Suggestions for Authors

Overall, your paper demonstrates a well-structured study of the relationship between food waste reduction, circular economy principles, and organizational theories in supply chains. Here are some critical feedback points for your consideration:

While you highlight the research gap and contribution, consider explicitly stating how your study addresses this gap. What unique insights or perspectives does your research offer that the current literature lacks?

Explicitly state the research objectives. This would provide a clear roadmap for readers and reviewers.

Expand on the review of existing organizational theories, especially those used in the operations management literature. Provide a critical analysis of how these theories have been applied in the context of food waste reduction.

Consider stating the criteria or rationale for selecting the six theories discussed in detail. This would add transparency to your approach.

Comments on the Quality of English Language

The language is appropriate.

Reviewer 3 Report

Comments and Suggestions for Authors

There is tremendous pressure on governments to maintain a balance between the supply and demand of food due to the ongoing global population growth and the depletion of both quantity and quality of resources. The Zero Hunger aim, which ranks second among the Sustainable Development Goals of the United Nations, emphasizes the same. Nonetheless, a large amount of food that is produced globally gets ruined or lost within the food supply chain.

In this context, the article addresses a topical issue and it is worth appreciating how well the theoretical framework is presented.

There are a multitude of theories that attempt to explain the behavior of organizations in reducing food waste such as: The Intermediary Actors Theory, Social Practice Theory, Perceived Behavioral Control, Food waste categorization theory, Institutional Theory, The Information System Theory, The Network Design Theory, Creating Shared Value Theory, Theory of Planned Behavior (TPB), etc.

Although I really liked the article from my point of view a number of improvements could be made, such as:

·        I did not identify the presentation of a quantitative or qualitative approach that would link the analyzed companies to the theoretical aspects. I did not understand how these connections were made.

·        How influential is one of the theories, or is it a group influence? How representative are the analyzed companies of the total population? Are the findings specific to a particular area and/or context?

·         I believe that an addition to the description for Figure 1 is needed.

·        On line 588, you mention something about the perspectives in figure 3. What is figure 3?

Round 2

Reviewer 1 Report

Comments and Suggestions for Authors

 Accepted in present form.

Reviewer 3 Report

Comments and Suggestions for Authors

Changes made by the authors as a result of the reviewers' comments improved the article.